**Data Availability Statement:** The clinical data cannot be shared publicly because of ethical and privacy shield restrictions. Researchers who meet the criteria for access to confidential data can apply for permission by contacting the Ethics Committee

# Trend in blood lead levels in Taiwanese adults 2005–2017

Chun-Wan Fang[1,2☯], Hsiao-Chen Ning[2,3☯], Ya-Ching Huang[1,2], Yu-Shao Chiang[4], Chun-Wei Chuang[1], I-Kuan Wang[5,6], Nai-Chia Fan[7,8], Cheng-Hao Weng[8,9], Wen-Hung Huang[8,9], Ching-Wei Hsu[8,9], Tzung-Hai Yen[8,9]*

1 Department of Laboratory Medicine, Chang Gung Memorial Hospital, Taoyuan, Linkou, Taiwan, 2 Department of Medical Biotechnology and Laboratory Science, College of Medicine, Chang Gung University, Taoyuan, Taiwan, 3 Department of Medical Research and Development, Chang Gung Memorial Hospital, Taoyuan, Linkou, Taiwan, 4 Department of Laboratory Medicine, Chang Gung Memorial Hospital, Kaohsiung, Taiwan, 5 Department of Nephrology, China Medical University Hospital, Taichung, Taiwan, 6 College of Medicine, China Medical University, Taichung, Taiwan, 7 Division of Pediatric General Medicine, Department of Pediatrics, Chang Gung Memorial Hospital, Linkou, Taiwan, 8 College of Medicine, Chang Gung University, Taoyuan, Taiwan, 9 Department of Nephrology, Clinical Poison Center, Kidney Research Center, Center for Tissue Engineering, Chang Gung Memorial Hospital, Linkou, Taiwan

☯ These authors contributed equally to this work.
* m19570@adm.cgmh.org.tw

## Abstract

This study examined the trend of blood lead levels (BLLs) in Taiwanese adults and analyzed the variations in the BLL between Linkou (northern) and Kaohsiung (southern) hospital branches. Between 2005 and 2017, 3,804 adult participants received blood lead tests at the Linkou (n = 2,674) and Kaohsiung (n = 1,130) branches of Chang Gung Memorial Hospital. The geometric mean of BLL was 2.77 µg/dL. The adult participants from the Kaohsiung branch were not only age older (49.8±14.1 versus 39.4±14.2 years; P<0.001) and male predominant (65.8 versus 41.7%; P<0.001) but also showed a higher BLL (4.45±3.93 versus 2.82±2.42 µg/dL; P<0.001) and lower estimated glomerular filtration rate (87.62±25.94 versus 93.67±23.88; P<0.001) than those from the Linkou branch. Multivariable logistic regression analysis revealed that the Kaohsiung branch [odds ratio (OR): 7.143; 95% confident interval (CI): 5.682–8.929; P<0.001], older age (OR: 1.008; 95% CI: 1.000–1.015; P = 0.043) and reduced estimated glomerular filtration rate (OR: 1.009; 95% CI: 1.004–1.014; P = 0.001) were significant predictors for BLL > 5 µg/dL. Therefore, this study confirmed a continuous decreasing trend in the BLL in Taiwan after banning leaded petrol in 2000.

## Introduction

Lead is known for its toxic effects on humans [1]. The symptoms of lead poisoning include impaired cognition, abdominal pain, irritability, impotence, insomnia, a metallic taste in the mouth and headache [2]. In advanced cases, it can lead to seizures, coma and death. The hematologic effects of lead intoxication are reduced hemoglobin synthesis and hemolysis. Lead may cause irreversible neurologic impairment, as well as chronic kidney disease, cardiovascular

of Chang Gung medical Foundation Institutional Review Board. Web address: https://www1.cgmh.org.tw/intr/intr1/c0040/. Postal address: B2F., No.123, Dinghu Rd., Guishan Dist., Taoyuan City 333, Taiwan. Tel: +8863 3 196200 extension 3705, 3707, 3708, 3709, 3711, 3712, 3713, 3716. Fax: +886 3 3494549.

**Funding:** This study was funded by research grants from Chang Gung Memorial Hospital (CORPG3K0192 and CMRPG3K2021). The funders had no role in study design, data collection and analysis, decision to publish, or preparation of the manuscript.

**Competing interests:** The authors have declared that no competing interests exist.

disease and reproductive damage [2]. Clinical research [3] has also shown that elevated blood lead levels (BLLs) are correlated with all-cause and cardiovascular-cause mortality in the general population.

In Taiwan, the phasing out of lead in petrol started in 1983, and the supply of leaded petrol was banned in 2000. Nevertheless, lead persists in the environment as a toxicant. The diagnosis of lead exposure is primarily based on an elevated BLL. Nevertheless, no level of lead is considered safe. A BLL > 3 μg/dL have been shown to produce reduced cognitive function and maladaptive behavior in many studies [4]. In 2006, Menke *et al.* [5] investigated U.S. adult participants with a BLL < 10 μg/dL and reported that the hazard ratios for comparisons of participants in the highest tertile of BLL ($\geq$ 3.62 μg/dL) with those in the lowest tertile (< 1.94 μg/dL) were 1.25 for all-cause mortality and 1.55 for cardiovascular mortality. The BLL was correlated with myocardial infarction and stroke mortality, and the correlation was apparent at levels of $\geq$ 2 μg/dL.

It has been two decades since the banning of leaded petrol in Taiwan. In USA [6], the geometric mean BLL dropped from 12.8 (1976–1980) to 0.82 μg/dL (2015–2016) after control of various lead sources in 1970. For Korea [7], the sale of leaded petrol was prohibited in 1993 and BLLs had decreased to < 2 μg/dL since 2000. However, there is little information regarding longitudinal trend of BLLs in Taiwan. Therefore, the objective of this study was to examine the trend in the BLLs in Taiwanese adult and to analyze the variations in the BLLs between the Southern and Northern parts of Taiwan.

## Results

Table 1 outlines the clinical data of the adult participants with an eGFR $\geq$ 60 mL/min/1.73 m$^2$, stratified by the Linkou or Kaohsiung branch. Between 2005 and 2017, 3,804 adult participants received blood lead tests at the Linkou (n = 2,674) or Kaohsiung (n = 1,130) branch of Chang Gung Memorial Hospital. The mean ages of the adult participants was 42.5 ± 14.9 years, and most were male (58.6%). The geometric mean BLL was 2.77 μg/dL. Adult participants from the Kaohsiung branch were not only age older (49.8 ± 14.1 versus 39.4 ± 14.2 years; P < 0.001)

**Table 1. Clinical data of patients, stratified by Linkou or Kaohsiung branch (n = 3804).**

| Variable | All adult participants (N = 3804) | Adult participants from Linkou branch of Chang Gung Memorial Hospital (n = 2674) | Adult participants from Kaohsiung branch of Chang Gung Memorial Hospital (n = 1130) | P value |
|---|---|---|---|---|
| Demographic data | | | | |
| Age (year) | 42.5 ± 14.9 | 39.4 ± 14.2 | 49.8 ± 14.1 | < 0.001*** |
| 18–60 years | 3269 (86.0) | 2383 (89.1) | 886 (78.4) | < 0.001*** |
| $\geq$ 60 years | 535 (14.0) | 291 (10.9) | 244 (21.6) | < 0.001*** |
| Male, n (%) | 2231 (58.6) | 1760 (65.8) | 471 (41.7) | < 0.001*** |
| Laboratory data | | | | |
| BLL (μg/dL) | 2.77 (2.68–2.86) | 2.45 (2.36–2.54) | 3.71 (3.48–3.94) | < 0.001*** |
| Serum creatinine level (mg/dL) | 0.86 ± 0.19 | 0.88 ± 0.20 | 0.83 ± 0.18 | < 0.001*** |
| eGFR (mL/min/1.73 m$^2$) | 91.88 ± 24.66 | 93.67 ± 23.88 | 87.62 ± 25.94 | < 0.001*** |

Note: BLL, blood lead level. eGFR, estimated glomerular filtration rate. BLL was expressed as geometric mean (range). Other parametric variables were presented as the means ± standard deviation, and nonparametric variables were presented as n (%). The geometric mean of BLL was used in this study because it was less affected by extreme values than the arithmetic mean. For comparisons between patient groups, Student's t test was used for parametric variables and Chi-square or Fisher's exact tests for nonparametric variables.

***P < 0.001.

and comprised more males (65.8 versus 41.7%; P < 0.001) but also showed a higher BLL (4.45 ± 3.93 versus 2.82 ± 2.42 μg/dL; P < 0.001) and lower eGFR (87.62 ± 25.94 versus 93.67 ± 23.88; P < 0.001) than adult participants from the Linkou branch. The participants are further divided into 18–60 years of age and 60 years and older. Nevertheless, there was no significant difference in BLL between participants with 18–60 years of age and 60 years and older (3.34 ± 3.15 versus 3.08 ± 2.22 μg/dL; P = 0.064).

A significant continuous decreasing trend in BLLs had been observed (P < 0.001, Fig 1). Moreover, there were rebounds of BLL noted at Kaoshiung branch of Chang Gung Memorial Hospital in 2009, 2013 and 2016. Rebounds of BLL were also noted at Linkou branch in 2009 and 2014. The highest mean BLL of the Kaohsiung branch appeared in 2005 (6.3 ± 4.9 μg/dL) and gradually decreased to 2.3 ± 1.3 μg/dL in 2017. The highest mean BLL of the Linkou branch appeared in 2008–2009 (3.5 ± 1.3 μg/dL) and gradually decreased to 1.8 ± 1.8 μg/dL in 2017.

Fig 2A shows the trend of adult participants with a BLL > 5 μg/dL. In total, the percentage of adult participants with a BLL > 5 μg/dL from the Kaohsiung branch was higher than that of the Linkou branch throughout these years. Fig 2B shows the trend of adult participants with a BLL > 10 μg/dL. The percentages of adult participants with a BLL > 10 μg/dL from the Kaohsiung branch were higher than those from the Linkou branch throughout the indicated years.

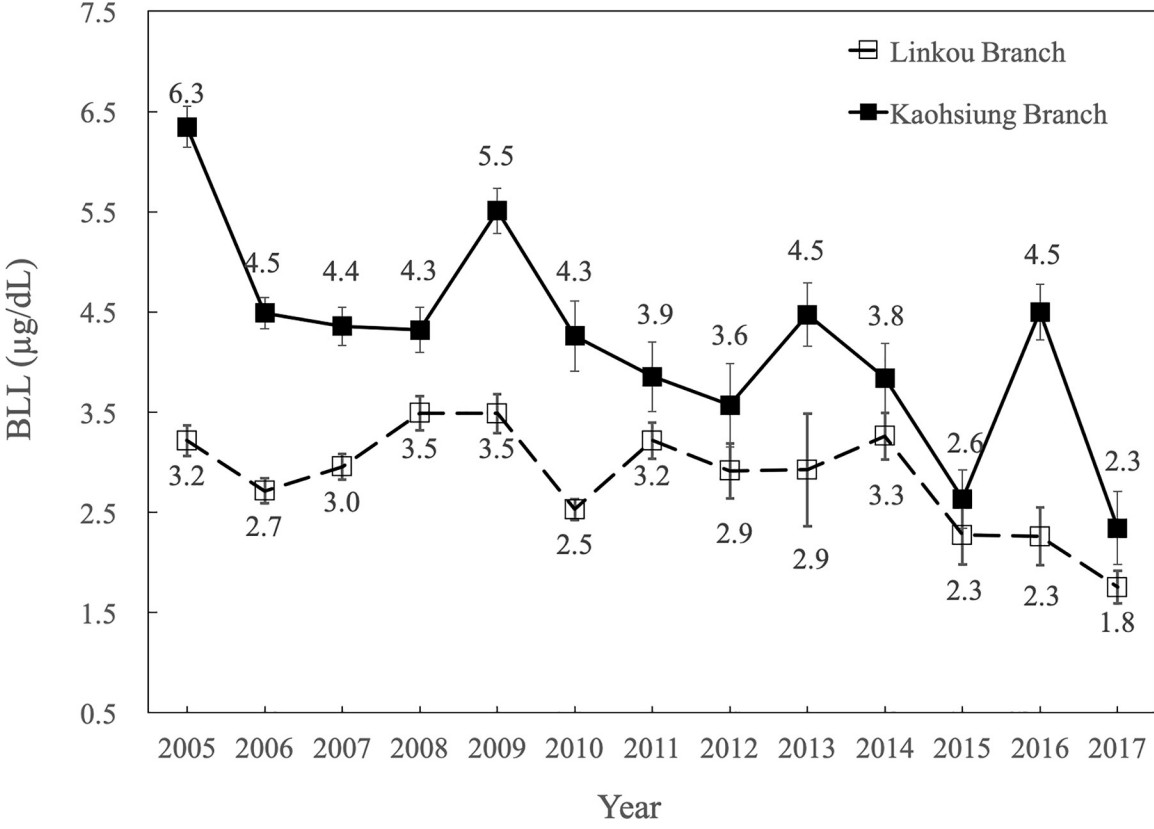

**Fig 1. Trend of blood lead levels (BLLs).** The figure shows the trend of BLLs in Taiwanese adult, 2005–2017. The differences between the means of BLLs in each year were examined by one-way analysis of variance test. A significant continuous decreasing trend (P < 0.001) was noted. Furthermore, the mean BLLs of the Kaohsiung branch were higher than those of the Linkou branch throughout the indicated periods [odds ratio (OR): 7.882; 95% CI (confident interval): 5.682–8.929].

**a**

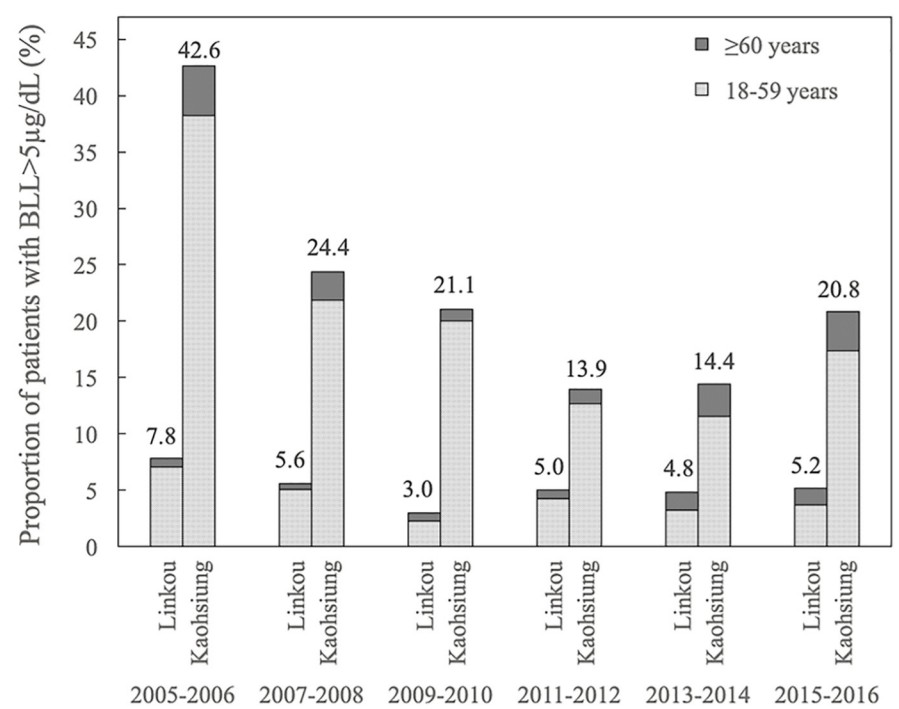

**b**

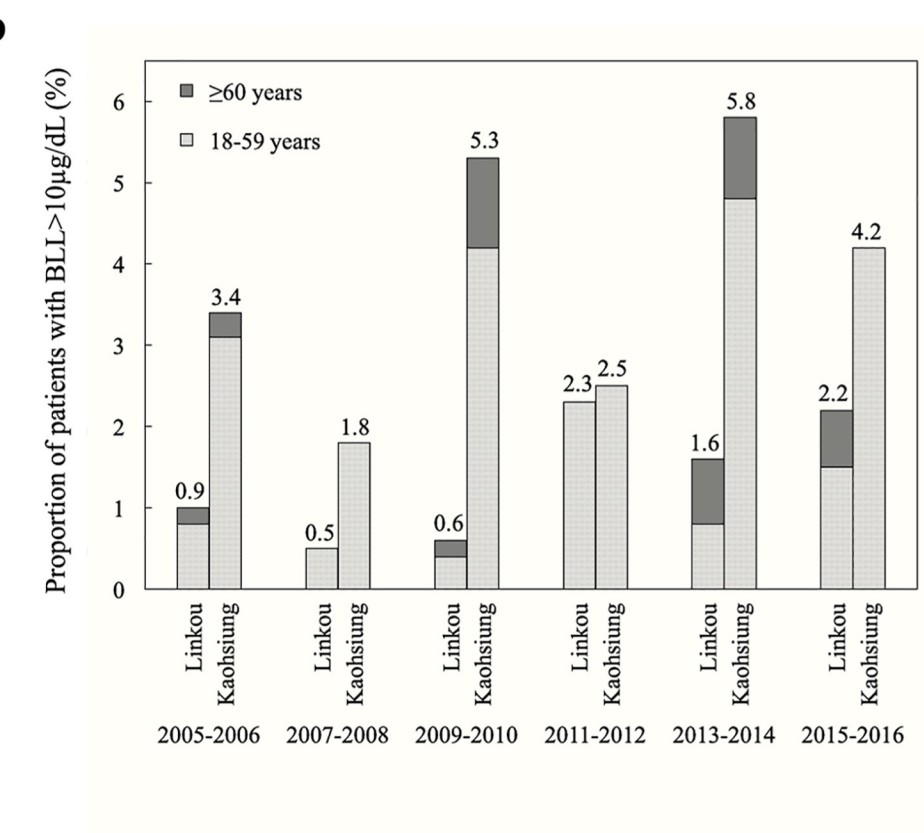

**Fig 2.** a. Trend of patients with blood lead levels (BLLs) > 5 μg/dL. b. Trend of patients with blood lead levels (BLLs) > 10 μg/dL.

Multivariable logistic regression analysis (Table 2) revealed that the Kaohsiung branch [odds ratio (OR): 7.882; 95% CI (confident interval): 5.682–8.929], male sex (OR: 1.523; 95% CI: 1.226–1.891) and a reduced eGFR (OR: 0.991; 95% CI: 0.986–0.996) were significant predictors for BLL > 5 μg/dL.

## Discussion

The data in the present study is the first from Taiwan in which both the BLL and eGFR were measured simultaneously, and the analysis confirmed the continuous decreasing trend in the BLL in both branches of Chang Gung Memorial Hospital (Fig 1).

As shown in Table 3 [8–20], a continuous decreasing trend exists in BLLs in Taiwan after banning leaded petrol in 2000. For example, in a study [8] conducted in 1989, Chiang and Chang reported that the mean BLL was higher for traffic policemen than for the control group (24.43 ± 5.31 μg/dL versus 20.14 ± 5.07 μg/dL, respectively; P < 0.01). The result is reasonable because traffic policemen are heavily exposed to vehicle exhaust of leaded petrol. However, another study [18] conducted in 2012 revealed that the mean BLL of children from southern Taiwan was higher than that of children from northern Taiwan (2.79 μg/dL versus 1.95 μg/dL, respectively; P < 0.001). Additionally, the boys' BLL was higher than the girls' BLL (2.52 μg/dL versus 2.14 μg/dL, respectively; P < 0.01).

A significant continuous decreasing trend in BLLs had been observed in our study cohort (Fig 1) and the geometric mean BLL was 2.77 μg/dL (Table 1). Moreover, there were rebounds of BLL noted at Kaoshiung branch of Chang Gung Memorial Hospital in 2009, 2013 and 2016. Rebounds of BLL were also noted at Linkou branch in 2009 and 2014. Nevertheless, there is no clear explanation. In 2016, Tsoi et al. [21] analyzed data from the National Health Nutrition and Examination Survey 1999–2014 and reported a continually decreased trend in BLL in American adult participants. The mean BLLs were 1.65 μg/dL, 1.44 μg/dL, 1.43 μg/dL, 1.29 μg/dL, 1.27 μg/dL, 1.12 μg/dL, 0.97 μg/dL, and 0.84 μg/dL in 1999–2000, 2001–2002, 2003–2004, 2005–2006, 2007–2008, 2009–2010, 2011–2012, and 2013–2014, respectively. In 2002, Becker et al. [22] analyzed data from the German Environmental Survey 1998 and reported that the geometric mean of BLL was 3.1 μg/dL. The SPECT-China Study [23] revealed that the median BLL was 4.40 μg/dL for men and 3.77 μg/dL for women in China. A Spanish study [24] found that the geometric mean BLL was 2.4 μg/dL in adult participants. A meta-analysis from India [25] noted a mean BLL of 7.52 μg/dL in non-occupationally exposed adult participants. The median BLL was 9.9 μg/dl in Congo adult participants [26]. Finally, the geometric means of BLL were 3.25 μg/dL and 6.60 μg/dL in non-occupationally exposed adult participants in Thailand [27] and Singapore [28], respectively.

**Table 2. Multivariable logistic regression analysis for risk factors associated with BLL > 5 μg/dL (n = 3804).**

| Variable | Odds ratio | 95% Confidence interval | P value |
|---|---|---|---|
| Kaohsiung branch of Chang Gung Memorial Hospital | 7.882 | 5.682–8.929 | < 0.001*** |
| Age (per 1 year increase) | 0.994 | 0.986–1.001 | 0.109 |
| Male sex | 1.523 | 1.226–1.891 | < 0.001*** |
| eGFR (per 1 mL/min/1.73 m² increase) | 0.991 | 0.986–0.996 | < 0.001*** |

Note: BLL, blood lead level. eGFR, estimated glomerular filtration rate.

***P < 0.001.

**Table 3. Published blood lead prevalence studies from Taiwan.**

| Study | Year | Sample size | Study group | Serum creatinine level (mg/dL) | eGFR (mL/min/1.73 m²) | BLL (µg/dL) |
|---|---|---|---|---|---|---|
| Chiang and Chang [8] | 1989 | 216 | Traffic policemen and students | Not mentioned | Not mentioned | 24.43 ± 5.31 (traffic policemen) |
| | | | | | | 20.14 ± 5.07 (students) |
| Liou et al. [9] | 1994 | 2919 | General population | Not mentioned | Not mentioned | 8.29 ± 5.92 |
| Liou et al. [10] | 1996 | 5913 | General population | Not mentioned | Not mentioned | 8.28 ± 5.39 |
| Liou et al. [11] | 1998 | 8828 | General population | Not mentioned | Not mentioned | 7.70 ± 5.23 |
| Chu et al. [12] | 1998 | 2803 | General population | Not mentioned | Not mentioned | 7.3±5.2 (male) |
| | | | | | | 5.7±3.9 (female) |
| Wang et al. [13] | 2002 | 934 | Primary school children | Not mentioned | Not mentioned | 5.50 ± 1.86 |
| Kuo et al. [14] | 2006 | 2565 | Aborigines | 1.0 ± 0.5 (aborigine) | Not mentioned | 5.6 ± 1.4 (aborigine male) |
| | | | | 1.0 ± 0.2 (non-aborigine) | | 5.4 ± 1.2 (aborigines female) |
| | | | | | | 5.3 ± 1.2 (non-aborigine male) |
| | | | | | | 5.3 ± 1.1 (non-aborigines female) |
| Chang et al. [15] | 2006 | 147 | Infertile women | Not mentioned | Not mentioned | 3.55 ± 1.39 (infertile women) 2.78 ± 2.05 (fertile women) |
| Yang et al. [16] | 2007 | 138 | Female nursing students | Not mentioned | Not mentioned | 2.64 ± 0.84 |
| Wu et al. [17] | 2009 | 154 | Immigrant women | Not mentioned | Not mentioned | 2.23 ± 1.63 (immigrant women) |
| | | | | | | 1.63 ± 1.00 (native women) |
| Huang et al. [18] | 2012 | 238 | Children | Not mentioned | Not mentioned | 2.79 ± 1.44 (Southern Taiwan) |
| | | | | | | 1.95 ± 1.51 (Northern Taiwan) |
| Hwang et al. [19] | 2014 | 934 | Preschool children | Not mentioned | Not mentioned | 1.84 ± 1.55 |
| Liu et al. [20] | 2017 | 40 | General population | Not mentioned | Not mentioned | 2.44 ± 0.97 |
| Current study | 2021 | 3804 | General population | 0.86 ± 0.19 | 91.88 ± 24.66 | 2.77 (2.68–2.86)[#] |

Note

[#]Data was expressed as geometric mean (range).

The major sources of body lead include ingestion (drinking water, paints, food and beverages) and inhalation (factory emission and automobile exhausts) [29]. As shown in Fig 1 and Table 1, adult participants from the Kaohsiung branch showed a higher BLL (4.45±3.93 versus 2.82±2.42 µg/dL; P < 0.001) than those from the Linkou branch. Previous data from our hospital [18] also revealed that the BLLs of pediatric participants from the Kaohsiung branch were higher than those of pediatric participants from the Linkou branch (2.79 versus 1.95 µg/dL, respectively; P < 0.001). This finding is not surprising because Kaohsiung is an international harbor and the heart of heavy and petrochemical industries in Taiwan. Incidents of occupational lead intoxication have been reported in Kaohsiung [30]. However, the Linkou branch is close to Taipei city and serves many residents from the Northern part of Taiwan. Taipei is a modern cosmopolitan city and is the political, economic, educational and cultural center of Taiwan. Therefore, industrial sources of lead are less frequent in Taipei city. Another possible explanation for the higher BLLs in the Kaohsiung branch is particulate matter 2.5 (PM2.5), which is often enriched with lead metal. According to the Environmental Protection Administration of Taiwan, Kaohsiung has been ranked as having the worst PM2.5 air pollution of any city in Taiwan in 2018. A previous study [31] reported that the ambient PM2.5 concentrations in Taipei and Kaohsiung were 23.09 and 48.47 µg/m³, respectively. Apart from air pollution, water pollution is a very serious problem in Kaohsiung city. For example, in July 2000, 100 tons of toxic solvents were dumped into the country's second longest river, Kaoping, leaving 3 million residents in and around Kaohsiung without drinking water for five days [32]. It is reasonable to assume that the industrial waste could contain toxic metals.

In this study, blood samples were analyzed for lead levels using two different methods, either GF-AAS or ICP-MS. In our hospital, the external quality control or test accuracy of both methods was maintained by participation in Proficiency Testing Program run by the College of American Pathologists. The ICP-MS technique is relatively simpler and faster, spending only one-tenth of the time necessary for GF-AAS with low detection limits [33]. Nevertheless, the accuracy and precision of the method are important in the laboratory analysis of BLLs. In 1997, Zhang *et al*. [33] demonstrated that the ICP-MS results could be mathematically corrected to be equivalent to the GF-AAS results. Both methods have been widely used as routine analytical methods in the determination of the lead level at our hospital.

Male gender was a significant predictor for BLL > 5 μg/dL (Table 3). This gender difference in BLLs was not surprising given that most past studies also revealed higher values of BLL in males than females. For example, the National Health Nutrition and Examination Survey 1999–2014 [21] indicated that BLLs were consistently higher in males than females. The German Environmental Survey 1998 [22] stated that geometric mean BLL was higher in males (3.6 μg/dL) than females (2.6 μg/dL). Study of general adult population in Iran [34] revealed that the reference values of BLLs for men and women were 16 μg/dL and 15 μg/dL, respectively. It was suggested [35] that the gender difference in BLLs could be explained by the fact that men were engaged in activities that may produce greater and more frequent exposure to lead.

Lead exposure, even at low-level, is associated with renal impairment and incident of chronic kidney disease. In a series of studies, it was demonstrated that environmental exposure to lead was related to progressive renal insufficiency in patients with [36] and without [37–39] diabetes, and chelation therapy may retard renal disease progression in these patients [40]. In a population-based study, researchers from Sweden found that low-level lead exposure was associated with reduced kidney function and incident chronic kidney disease [41]. In another cross-sectional study, researchers from United States [42] reported that low-level lead exposure was associated with lower intelligence quotient and more inattention in children with chronic kidney disease. In a recent systematic review and meta-analysis study [43], it was confirmed that lead exposure was associated with reduced estimated glomerular filtration rate and increased proteinuria risks.

As stated by World Health Organization [1], primary prevention (elimination of exposure to lead at its source) is the single most effective intervention against lead poisoning. In addition to phasing out use of leaded petrol, Taiwan government has been taking many actions to eliminate the use of lead in many products. There is no official guideline for lead exposure in Taiwan, but a whole blood lead test is the test of choice for the diagnosis and monitoring of lead poisoning. It is recommended that BLL should be < 10 μg/dL but < 5 μg/dL for children aged 1–5 years.

Since this study involves retrospective analysis of existing data, only adult participants who visited Chang Gung Memorial Hospital between 2005 and 2017 were recruited for analysis. Therefore, this study is clearly limited by the lacking of standard sampling protocol to select study participants. Furthermore, the study participants were recruited from hospital, which might differ from general population. Further epidemiologic studies with standard sampling protocol targeting at recruiting participants from general population are warranted.

## Conclusion

In summary, the geometric mean BLL for Taiwanese adult was 2.77 μg/dL. Adult participants from the Kaohsiung branch were not only age older and male predominant but also showed higher BLLs and had lower estimated glomerular filtration rate than adult participants from

the Linkou branch. The Kaohsiung branch, male sex and a reduced estimated glomerular filtration rate were risk factors associated with a BLL > 5 μg/dL. Finally, this study confirmed a continuous decreasing trend in BLLs in Taiwan after the banning of leaded petrol in 2000.

## Materials and methods

### Ethical statement

This observational study adhered to the guidelines of Declaration of Helsinki and was approved by the Medical Ethics Committee of Chang Gung Memorial Hospital. Since this study included a retrospective analysis of existing data, Institutional Review Board approval was acquired without specific informed consent from the patients. The Institutional Review Board of the Chang Gung Memorial Hospital had waived the need for consent, and the Institutional Review Board number was 201701031A3.

### Participants

This study was conducted in adult participants who visited the Linkou or Kaohsiung branches of Chang Gung Memorial Hospital and received their blood lead tests between 2005 and 2017. Since these patients also received blood test for creatinine, the estimated glomerular filtration rate was a calculation based on a serum creatinine test. The blood lead level was only measured once per subject. Chang Gung Memorial Hospital offers the largest health care services in Taiwan, comprising a network of several hospital branches located in Linkou, Taipei, Taoyuan, Keelung, Yunlin, Chiayi, and Kaohsiung (Web address: http://www.chang-gung.com/en/about.aspx?id=11&bid=1). The Linkou and Kaohsiung branches are the two main branches. The Linkou branch primarily serves patients from the northern part of Taiwan, while the Kaohsiung branch primarily serves patients from the southern part of Taiwan.

### Inclusion and exclusion criteria

All the adult participants with an estimated glomerular filtration rate $\geq$ 60 mL/min/1.73 m$^2$ and who received blood lead tests between 2005 and 2017 were recruited for this study. Adult participants with an estimated glomerular filtration rate < 60 mL/min/1.73 m$^2$ or participants aged younger than 18 years were excluded from the analysis. Theoretically, lead is concentrated in the kidney after body absorption and is then excreted in urine [44]. Patients with chronic kidney disease might show elevated BLLs due to decreased urinary excretion of lead. Therefore, participants with estimated glomerular filtration rate < 60 mL/min/1.73 m$^2$ were excluded because this could confound the analyses.

### Blood sample collection and serum biochemistry

Whole blood samples from each participant were collected in metal-free tubes (Vacutainer; BD 368381; Becton-Dickson, Franklin Lakes, NJ, USA). The BLLs were measured using either graphite furnace atomic absorption spectrometry (GFAAS) (PinAAcle 900T; PerkinElmer, Waltham, Massachusetts, USA) or inductively coupled plasma mass spectrometry (ELAN DRC-e; PerkinElmer, Waltham, Massachusetts, USA). The serum creatinine levels were measured using an automated biochemical analyzer (LABOSPECT 008; Hitachi, Tokyo, Japan).

### Graphite furnace atomic absorption spectrophotometry (GF-AAS)

Whole blood specimens (200 μL) were diluted (1 + 4) with a matrix modifier solution. All the standards were purchased from High-Purity Standards, traceable to the NIST (South Carolina, USA). Calibration was performed using a reagent blank with five calibration standards.

Calibration curves for lead had an R ≥ 0.995 and a blank absorbance < 0.005. The recovery rate for lead was within 90%–110%. The limit of quantification was 1.0 μg/dL. Quality controls were analyzed at the start and end of each analytical run, and once per level again after every 10 samples. Test precision was monitored using a coefficient of variation of less than 5% at each level of control. The test accuracy was validated regularly by the College of American Pathologist proficiency testing.

## Inductively coupled plasma mass spectrometry (ICP-MS)

Whole blood specimens (250 μL) were diluted (1 + 20) with 1.5% nitric acid (JT Baker, New Jersey, USA) solution containing yttrium as an internal standard. All the standards were purchased from High-Purity Standards (South Carolina, USA). Calibration was performed using a reagent blank and six calibration standards in the internal standard diluent solution. Calibration curves for lead had an R ≥ 0.995. The recovery rate for lead was 90%~110%. The limit of quantification was 0.7 μg/dL. Quality controls were analyzed at the start and end of each analytical run, and once per level again after every 10 samples. Test precision was monitored using a coefficient of variation less than 5% at each level of control. The test accuracy was validated regularly by the College of American Pathologist proficiency testing.

## Statistical analysis

Statistical analysis was performed using PASW software (version 18.0; SPSS Inc., Chicago, IL, USA). Parametric variables were expressed as the means and standard deviations, and nonparametric variables were expressed as numbers with percentages in brackets. All the data were tested for normality of distribution and equality of standard deviations before analysis. The Quantile-quantile plot and Kolmogorov-Smirnov test were used to check the normality of distribution. The Levene test was used to check the equality of variance. For comparisons between patient groups, we used Student's t test for parametric variables and Chi-square or Fisher's exact tests for nonparametric variables. The differences between the means of BLLs in each year were examined by one-way analysis of variance test. A multivariable binary logistic regression analysis was performed to analyze potential predictors for BLL > 5 μg/dL. The variables included age, sex, estimated glomerular filtration rate and branch of Chang Gung Memorial Hospital. A P value < 0.05 was considered statistically significant.

## Acknowledgments

Parts of this study were presented at the 16th International Congress of Therapeutic Drug Monitoring & Clinical Toxicology held from 16th to 19th September 2018 in Brisbane, Australia.

## Author Contributions

**Conceptualization:** Chun-Wan Fang, Hsiao-Chen Ning.

**Data curation:** Chun-Wan Fang, Hsiao-Chen Ning.

**Formal analysis:** Ya-Ching Huang, Yu-Shao Chiang, Chun-Wei Chuang, I-Kuan Wang, Nai-Chia Fan.

**Resources:** Cheng-Hao Weng, Wen-Hung Huang, Ching-Wei Hsu.

**Supervision:** Tzung-Hai Yen.

**Writing – original draft:** Chun-Wan Fang, Hsiao-Chen Ning.

**Writing – review & editing:** Tzung-Hai Yen.

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
