## [Decision Letter · Decision Letter 0]

17 Sep 2021

PONE-D-21-24583

Trend in blood lead levels in Taiwanese adults 2005 - 2017

PLOS ONE

Dear Dr. Tzung-Hai Yen, 

Thank you for submitting your manuscript to PLOS ONE. After careful consideration, we feel that it has merit but does not fully meet PLOS ONE’s publication criteria as it currently stands. Therefore, we invite you to submit a revised version of the manuscript that addresses the points raised during the review process.

Please give a point to point reply of the reviewer comments in your revised manuscript.

We look forward to receiving your revised manuscript.

Kind regards,

Purvi Purohit

Academic Editor

PLOS ONE

Journal Requirements:

Additional Editor Comments:

The authors are required to do some minor revisions throughout the manuscript as pointed out by the two reviewers.

Please ensure that point to point explanation of the reviewer comments is made while submitting the revised manuscript.

Reviewers' comments:

Reviewer's Responses to Questions

**Comments to the Author**

1. Is the manuscript technically sound, and do the data support the conclusions?

Reviewer #1: Yes

Reviewer #2: Yes

2. Has the statistical analysis been performed appropriately and rigorously? 

Reviewer #1: Yes

Reviewer #2: Yes

3. Have the authors made all data underlying the findings in their manuscript fully available?

Reviewer #1: No

Reviewer #2: No

4. Is the manuscript presented in an intelligible fashion and written in standard English?

Reviewer #1: Yes

Reviewer #2: Yes

5. Review Comments to the Author

Reviewer #1: The authors here present a trend in blood lead levels in Taiwanese adults. The objectives, methods, and summary of the results have been described succinctly with relevant statistics. However, minor revisions need to be incorporated.

General:

1. There are syntax and punctuation issues throughout the document, which detracts the readability of the text. I have highlighted a few, but a thorough proofreading of the manuscript would be beneficial. Some of the longer sentences could also benefit from rephrasing. Further, a period should go after "et al" regardless of where it is mentioned in the sentence.

2. It would be helpful for the readers if the statistics/tests performed for the comparisons in the tables and figures are also mentioned in the table footnotes/figure legends.

Abstract:

The last sentence is a repetition of the findings already mentioned, and hence, can be removed.

Introduction:,.

Line 66: remove ‘because’

Results:

Line 85: age

Table 1 depicts the demographic and laboratory data. In the demographic data, the participants are further divided into 18-60 years of age and 60 years and older.

(1) Both groups seem to have included the participants who are 60 years of age. Based on the data shown in Figure 2a and 2b, I presume the first group is 18-59 years and the second group would be 60 years and older. Kindly rectify.

(2) In Table 1, the BLL should also be compared between these two age groups separately.

(3) The BLL is reported as geometric mean and range in the tables 1 and 3 but as geometric mean and SD (this is geometric SD, I presume; kindly clarify) elsewhere in the text. The authors should address the reason for this or keep the reporting uniform throughout.

Line 97: The authors mention a significant decreasing trend of BLL. Statistical details of how the differences between both the trends were analyzed can be included in the figure legend (Figure 1).

Lines 98-99, 145-156: BLL rebounds are noted in Linkou branch in 2009 and 2016. However, Figure 1 seems to depict the same for Kaosiung branch. Kindly confirm or address.

Line 135: Fix the citation.

Discussion:

Since eGFR was a significant predictor for blood lead levels, a brief section can be added discussing the relation between lead exposure and kidney function citing relevant literature.

Lines 155, 157: same word with and without hyphen; make them uniform.

Line 198: “by the fact..”

Line 210: “by the lack of a standard sampling protocol..”

Methods:

In the statistics section, mention the tests used for checking the normality of distribution and equality of variance.

Reviewer #2: Manuscript is technically sound, and the data supports the conclusions. The statistical analysis has been performed appropriately.The data points behind means, medians and variance measures should be made available.

1. In introduction section: Please elaborate on "BLL < 10 μg/dL showed higher cardiovascular mortality than those with BLLs >3.61 μg/dL and <10 μg/dL"...

2. In results section"Figure 1 Trend of blood lead levels (BLLs) in Taiwanese adult, 2005 -2017" should be rephrased to Figure 1 shows the trend of blood lead levels (BLLs) in Taiwanese adult, 2005 -2017.... and CI (confident interval): 5.682 – 8.929]"...its confidence interval

3. In discussion section the sentence " It is recommended that BLL should be < 10 µg/dL but < 5 µg/dL for children aged 1–5 years" should be elaborated. eGFR is the novelty of the paper and hence BLL and eGFR should be discussed in detail.

4. In methodology, inclusion exclusion criteria "Lead is concentrated in the kidney after body absorption and is then excreted in urine. Patients with chronic kidney disease might show elevated BLLs due to decreased urinary excretion"....this is a generalized statement and support the statement with adequate references.

6. PLOS authors have the option to publish the peer review history of their article (what does this mean?). If published, this will include your full peer review and any attached files.

Reviewer #1: No

Reviewer #2: No

---

## [Author Response · Author response to Decision Letter 0]

31 Oct 2021

Reviewer #1: The authors here present a trend in blood lead levels in Taiwanese adults. The objectives, methods, and summary of the results have been described succinctly with relevant statistics. However, minor revisions need to be incorporated.

General:

1. There are syntax and punctuation issues throughout the document, which detracts the readability of the text. I have highlighted a few, but a thorough proofreading of the manuscript would be beneficial. Some of the longer sentences could also benefit from rephrasing. Further, a period should go after "et al" regardless of where it is mentioned in the sentence.

Response: Thank you for the comments. This manuscript has been sent to AJE for English grammar editing prior to first submission. Please advise us again if we miss out anything.

2. It would be helpful for the readers if the statistics/tests performed for the comparisons in the tables and figures are also mentioned in the table footnotes/figure legends.

Response: Thank you for the comment. The information has been included in the table footnotes/figure legends.

Abstract:

The last sentence is a repetition of the findings already mentioned, and hence, can be removed.

Response: Thank you for the comment. The sentence has been removed.

Introduction:,.

Line 66: remove ‘because’

Response: Thank you for the comment. The word has been removed.

Results:

Line 85: age

Response: Thank you for the comment. The word has been included.

Table 1 depicts the demographic and laboratory data. In the demographic data, the participants are further divided into 18-60 years of age and 60 years and older.

(1) Both groups seem to have included the participants who are 60 years of age. Based on the data shown in Figure 2a and 2b, I presume the first group is 18-59 years and the second group would be 60 years and older. Kindly rectify.

Response: Thank you for the comment. The paragraph has been revised.

The participants are further divided into 18 - 60 years of age and 60 years and older. Nevertheless, there was no significant difference in BLL between participants with 18 - 60 years of age and 60 years and older (3.34 ± 3.15 versus 3.08 ± 2.22 µg/dL; P = 0.064).

(2) In Table 1, the BLL should also be compared between these two age groups separately.

Response: Thank you for the comment. The paragraph has been revised.

The participants are further divided into 18-60 years of age and 60 years and older. Nevertheless, there was no significant difference in BLL between participants with 18 - 60 years of age and 60 years and older (3.34 ± 3.15 versus 3.08 ± 2.22 µg/dL; P = 0.064).

(3) The BLL is reported as geometric mean and range in the tables 1 and 3 but as geometric mean and SD (this is geometric SD, I presume; kindly clarify) elsewhere in the text. The authors should address the reason for this or keep the reporting uniform throughout.

Response: Thank you for the comment. The footnote of Table 1 and 3 have been revised.

BLL was expressed as geometric mean (range). Other parametric variables were presented as the means ± standard deviation, and nonparametric variables were presented as n (%). The geometric mean of BLL was used in this study because it was less affected by extreme values than the arithmetic mean.

Line 97: The authors mention a significant decreasing trend of BLL. Statistical details of how the differences between both the trends were analyzed can be included in the figure legend (Figure 1).

Response: Thank you for the comment. The figure legend has been revised according to comments of both reviewers.

Figure 1 Trend of blood lead levels (BLLs). The figure shows the trend of BLLs in Taiwanese adult, 2005 -2017. The differences between the means of BLLs in each year were examined by one-way analysis of variance test. A significant continuous decreasing trend (P < 0.001) was noted. Furthermore, the mean BLLs of the Kaohsiung branch were higher than those of the Linkou branch throughout the indicated periods [odds ratio (OR): 7.882; 95% CI (confident interval): 5.682 – 8.929].

Lines 98-99, 145-156: BLL rebounds are noted in Linkou branch in 2009 and 2016. However, Figure 1 seems to depict the same for Kaosiung branch. Kindly confirm or address.

Response: Thank you for the comment. The sentences have been revised. 

A significant continuous decreasing trend in BLLs had been observed (P < 0.001, Figure 1). Moreover, there were rebounds of BLL noted at Kaoshiung branch of Chang Gung Memorial Hospital in 2009, 2013 and 2016. Rebounds of BLL were also noted at Linkou branch in 2009 and 2014.

Line 135: Fix the citation.

Response: Thank you for reminding us. The citation has been revised.

Discussion:

Since eGFR was a significant predictor for blood lead levels, a brief section can be added discussing the relation between lead exposure and kidney function citing relevant literature.

Response: Thank you for the comment. A new paragraph has been added in Discussion section.

Lead exposure, even at low-level, is associated with renal impairment and incident of chronic kidney disease. In a series of studies, it was demonstrated that environmental exposure to lead was related to progressive renal insufficiency in patients with [36] and without [37-39] diabetes, and chelation therapy may retard renal disease progression in these patients [40]. In a population-based study, researchers from Sweden found that low-level lead exposure was associated with reduced kidney function and incident chronic kidney disease [41]. In another cross-sectional study, researchers from United States [42] reported that low-level lead exposure was associated with lower intelligence quotient and more inattention in children with chronic kidney disease. In a recent systematic review and meta-analysis study [43], it was confirmed that lead exposure was associated with reduced estimated glomerular filtration rate and increased proteinuria risks. 

Lines 155, 157: same word with and without hyphen; make them uniform.

Response: Thank you for the comment. The word has been revised.

Line 198: “by the fact..”

Response: Thank you for the comment. The sentence has been revised.

Line 210: “by the lack of a standard sampling protocol..”

Response: Thank you for the comment. The sentence has been revised.

Methods:

In the statistics section, mention the tests used for checking the normality of distribution and equality of variance.

Response: Thank you for the comment. The analytical method has been provided.

The Quantile-quantile plot and Kolmogorov-Smirnov test were used to check the normality of distribution. The Levene test was used to check the equality of variance. 

Reviewer #2: Manuscript is technically sound, and the data supports the conclusions. The statistical analysis has been performed appropriately. The data points behind means, medians and variance measures should be made available.

1. In introduction section: Please elaborate on "BLL < 10 μg/dL showed higher cardiovascular mortality than those with BLLs >3.61 μg/dL and <10 μg/dL"...

Response: Thank you for the comment. The study has been elaborated. 

In 2006, Menke et al [5] investigated U.S. adult participants with a BLL < 10 μg/dL and reported that the hazard ratios for comparisons of participants in the highest tertile of BLL (≥ 3.62 μg/dL) with those in the lowest tertile (< 1.94 μg/dL) were 1.25 for all-cause mortality and 1.55 for cardiovascular mortality. The BLL was strongly correlated with myocardial infarction and stroke mortality, and the correlation was apparent at levels of ≥ 2 μg/dL.

2. In results section "Figure 1 Trend of blood lead levels (BLLs) in Taiwanese adult, 2005 -2017" should be rephrased to Figure 1 shows the trend of blood lead levels (BLLs) in Taiwanese adult, 2005 -2017.... and CI (confident interval): 5.682 – 8.929]"...its confidence interval

Response: Thank you for the comment. The figure legend has been revised according to comments of both reviewers.

Figure 1 Trend of blood lead levels (BLLs). The figure shows the trend of BLLs in Taiwanese adult, 2005 -2017. The differences between the means of BLLs in each year were examined by one-way analysis of variance test. A significant continuous decreasing trend (P < 0.001) was noted. Furthermore, the mean BLLs of the Kaohsiung branch were higher than those of the Linkou branch throughout the indicated periods [odds ratio (OR): 7.882; 95% CI (confident interval): 5.682 – 8.929].

3. In discussion section the sentence " It is recommended that BLL should be < 10 µg/dL but < 5 µg/dL for children aged 1–5 years" should be elaborated. eGFR is the novelty of the paper and hence BLL and eGFR should be discussed in detail.

Response: Thank you for the comment. A new paragraph has been added in Discussion section.

Lead exposure, even at low-level, is associated with renal impairment and incident of chronic kidney disease. In a series of studies, it was demonstrated that environmental exposure to lead was related to progressive renal insufficiency in patients with [36] and without [37-39] diabetes, and chelation therapy may retard renal disease progression in these patients [40]. In a population-based study, researchers from Sweden found that low-level lead exposure was associated with reduced kidney function and incident chronic kidney disease [41]. In another cross-sectional study, researchers from United States [42] reported that low-level lead exposure was associated with lower intelligence quotient and more inattention in children with chronic kidney disease. In a recent systematic review and meta-analysis study [43], it was confirmed that lead exposure was associated with reduced estimated glomerular filtration rate and increased proteinuria risks. 

4. In methodology, inclusion exclusion criteria "Lead is concentrated in the kidney after body absorption and is then excreted in urine. Patients with chronic kidney disease might show elevated BLLs due to decreased urinary excretion"....this is a generalized statement and support the statement with adequate references.

Response: Thank you for the comment. The statement has been referenced.

---

## [Editor Report · Decision Letter 1]

17 Nov 2021

Trend in blood lead levels in Taiwanese adults 2005 - 2017

PONE-D-21-24583R1

Dear Dr. Tzung-Hai Yen

We’re pleased to inform you that your manuscript has been judged scientifically suitable for publication and will be formally accepted for publication once it meets all outstanding technical requirements.

Kind regards,

Purvi Purohit

Academic Editor

PLOS ONE

Additional Editor Comments (optional):

The authors have responded appropriately to all the queries raised by the reviewers. The manuscript has been revised and the manuscript is now acceptable for publication.

---

## [Editor Report · Acceptance letter]

22 Nov 2021

PONE-D-21-24583R1 

Trend in blood lead levels in Taiwanese adults 2005 - 2017 

Dear Dr. Yen:

I'm pleased to inform you that your manuscript has been deemed suitable for publication in PLOS ONE. Congratulations! Your manuscript is now with our production department. 

Kind regards, 

on behalf of

Dr. Purvi Purohit 

Academic Editor

PLOS ONE